# Renal Hypouricemia 1: Rare Disorder as Common Disease in Eastern Slovakia Roma Population

**DOI:** 10.3390/biomedicines9111607

**Published:** 2021-11-03

**Authors:** Blanka Stiburkova, Jana Bohatá, Kateřina Pavelcová, Velibor Tasic, Dijana Plaseska-Karanfilska, Sung-Kweon Cho, Ludmila Potočnaková, Jana Šaligová

**Affiliations:** 1Institute of Rheumatology, 128 00 Prague, Czech Republic; bohata@revma.cz (J.B.); pavelcova@revma.cz (K.P.); 2Department of Pediatrics and Inherited Metabolic Disorders, First Faculty of Medicine, Charles University, General University Hospital, 121 00 Prague, Czech Republic; 3Department of Rheumatology, First Faculty of Medicine, Charles University, 121 08 Prague, Czech Republic; 4Faculty of Medicine, University Ss. Cyril and Methodius, 1000 Skopje, North Macedonia; vtasic2003@gmail.com; 5Research Centre for Genetic Engineering and Biotechnology “Georgi D. Efremov”, Macedonian Academy of Sciences and Arts, 1000 Skopje, North Macedonia; dijana@manu.edu.mk; 6Department of Pharmacology, Ajou University School of Medicine, 164, Worldcup-ro, Yeongtong-gu, Suwon 16499, Korea; wontan@ajou.ac.kr; 7Metabolic Ambulance of Department of Paediatrics and Adolescent Medicine, Children’s Faculty Hospital, 040 11 Košice, Slovakia; lpotocnakova@gmail.com (L.P.); jana.saligova@gmail.com (J.Š.)

**Keywords:** renal hypouricemia, *SLC22A12*, URAT1, ethnic specificity, Roma

## Abstract

Renal hypouricemia (RHUC) is caused by an inherited defect in the main reabsorption system of uric acid, *SLC22A12* (URAT1) and *SLC2A9* (GLUT9). RHUC is characterized by a decreased serum uric acid concentration and an increase in its excreted fraction. Patients suffer from hypouricemia, hyperuricosuria, urolithiasis, and even acute kidney injury. We report clinical, biochemical, and genetic findings in a cohort recruited from the Košice region of Slovakia consisting of 27 subjects with hypouricemia and relatives from 11 families, 10 of whom were of Roma ethnicity. We amplified, directly sequenced, and analyzed all coding regions and exon–intron boundaries of the *SLC22A12* and *SLC2A9* genes. Sequence analysis identified dysfunctional variants c.1245_1253del and c.1400C>T in the *SLC22A12* gene, but no other causal allelic variants were found. One heterozygote and one homozygote for c.1245_1253del, nine heterozygotes and one homozygote for c.1400C>T, and two compound heterozygotes for c.1400C>T and c.1245_1253del were found in a total of 14 subjects. Our result confirms the prevalence of dysfunctional URAT1 variants in Roma subjects based on analyses in Slovak, Czech, and Spanish cohorts, and for the first time in a Macedonian Roma cohort. Although RHUC1 is a rare inherited disease, the frequency of URAT1-associated variants indicates that this disease is underdiagnosed. Our findings illustrate that there are common dysfunctional URAT1 allelic variants in the general Roma population that should be routinely considered in clinical practice as part of the diagnosis of Roma patients with hypouricemia and hyperuricosuria exhibiting clinical signs such as urolithiasis, nephrolithiasis, and acute kidney injury.

## 1. Introduction

Hypouricemia, defined as a serum uric acid (UA) concentration < 120 µmol/L, is a rare laboratory finding with a prevalence of 0.21–0.53% that varies with age [1,2]. Secondary hypouricemia may be caused by Fanconi syndrome, cystinosis, diabetes mellitus, and/or inappropriate antidiuretic hormone secretion syndrome. Primary hypouricemia can result from decreased UA production by a blockade of the last step of purine degradation, e.g., hereditary xanthinuria, but is more commonly due to decreased renal tubular UA reabsorption, e.g., renal hypouricemia.

Renal hypouricemia (RHUC) is a rare heterogeneous inherited disease caused by a dysfunction of the primary renal urate transporters: URAT1 (gene *SLC22A12*, OMIM #220150) and GLUT9 (gene *SLC2A9*, OMIM #612076), resulting in impaired tubular transport of UA, insufficient reabsorption, and/or increased secretion by a mechanism of endothelial dysfunction [3,4,5,6,7]. The biochemical markers are characterized by hypouricemia and an increased fractional excretion of UA (FE-UA). Clinically, the disease presents with urolithiasis, nephrolithiasis, and, in some patients, acute kidney injury that often occurs after physical exertion [8]. No treatment is available; however, allopurinol has been used to prevent the recurrence of acute kidney injury episodes [9].

RHUC patients have been described in different ethnic groups and geographically noncontiguous countries; over 200 cases with URAT1 variants and ~20 cases with GLUT9 defects have been reported to date worldwide. The loss-of-function *SLC22A12* variants (in compound heterozygous and/or homozygous state) are the primary cause of renal hypouricemia (> 90%), with significant population specificity: the high incidence of RHUC1 in Japanese and Korean patients reflects the 2.3% allelic frequency of the prevalent dysfunctional variant rs121907892 (p.W258X) [10], with null allele frequency in African/American, Ashkenazi Jewish, South Asia, and European populations (130,978 subjects, ExAc database). However, the world’s highest frequency of predominant dysfunctional RHUC1 variants was identified in the European Roma population (1016 individuals from regions of the Czech Republic, Slovakia, and Spain): the rs200104135 (p.T467M) variant has a frequency of 5.6%, and the deletion variant p.L415_G417 has a frequency of 1.9% [11].

In the present study, we collected a cohort of pediatric subjects with repeated findings of serum uric acid below 120 µmol/L. A cohort of 27 subjects was recruited as part of a single-center study by the Metabolic Ambulance of the Department of Paediatrics and Adolescent Medicine, Children’s Faculty Hospital, Košice, Slovakia, between 2018 and 2020 and consisted of probands and relatives from 11 families. The aims of the study were (a) to determine whether renal hypouricemia type 1 and 2 occurs in the cohort, (b) to determine which type of renal hypouricemia predominates, and (c) to identify causal variants for this rare Mendelian disease of urate transport associated with hypouricemia.

## 2. Materials and Methods

### 2.1. Clinical Subjects

The studied probands and their family members were Slovaks of Roma ethnicity, except for family I, who did not report their ethnicity. Participants were repeatedly diagnosed with serum uric acid below 120 µmol/L in children and below 150 µmol/L in adults during a two-year study period. The families under investigation did not indicate a family relationship. All participants were fully informed of the study’s goals and written informed consent was obtained from each participant before enrollment in the study. All tests were performed following standards set by institutional ethics committees. The study was approved on 24 July 2018 as part of project no. 7131/2018. All the procedures were performed per the Declaration of Helsinki. A cohort of genomic DNA of 109 unrelated subjects of Roma ethnicity (chosen irrespective of their state of health) from Macedonia and a previously reported cohort of 1016 Roma individuals from five European regions (Slovakia, Czech Republic, Spain) was used as a control group [11,12].

### 2.2. Clinical Investigations and Sequence Analyses

Urate and creatinine levels were measured using specific enzymatic methods, and the Jaffé reaction, which was adapted for use with an auto-analyzer (Hitachi Automatic Analyzer 902; Roche, Basel, Switzerland). Whole blood was collected and stored in the biobank of the Institute of Rheumatology, Prague, Czech Republic. Subsequently, genomic DNA was extracted using QIAamp DNA Mini Kits (QiagenGmbH, Hilden, Germany). All coding exons and intron–exon boundaries of *SLC22A12* and *SLC2A9* were amplified from genomic DNA using polymerase chain reaction and subsequently purified using PCR DNA Fragments Extraction Kits (Geneaid, New Taipei City, Taiwan). DNA sequencing was performed on an automated 3130 Genetic Analyzer (Applied Biosystems Inc., Foster City, CA, USA). Primer sequences and PCR conditions used for amplification were previously described [12,13,14].

The reference sequences for GLUT9/*SLC2A9* (NM_020041.2; NP_064425.2; SNP source dbSNP 132) and URAT1/*SLC22A12* (NM_144585.3) were defined as version NC_000004.12 (Chromosome 4: 9,771,153–10,054,936) and NC_000011.8 (Chromosome 11:64,114,688–64,126,396), respectively.

In the control group of 109 unrelated subjects of Roma ethnicity from Macedonia, we only analyzed exon 7 and exon 9 of *SLC22A12* for identifying prevalent variants c.1245_1253del (p.L415_G417del) and c.1400C>T (rs200104135, p.T467M).

## 3. Results

### 3.1. Clinical Subjects

The cohort consisted of nine hypouricemia probands and 18 family relatives. The probands had repeated hypouricemia and/or increased FE-UA. Hypouricemia was the determining factor for inclusion in the study, not ethnicity; however, all subjects were of Roma origin, except family I, who did not report their ethnicity. No clinical or laboratory symptoms of renal disease were present in the probands. However, other hereditary disorders were present in the cohort: five subjects were diagnosed with phenylketonuria (PKU, OMIM #261600), four subjects with short-chain acyl-CoA dehydrogenase deficiency (SCADD, OMIM #201470), one subject with Gilbert syndrome (OMIM #143500), one subject with congenital hypothyroidism (OMIM #275200), and one subject with mild hyperhomocysteinemia due to a methylenetetrahydrofolate reductase deficiency (OMIM #607093.0004). The clinical and biochemical data from this study are summarized in Table 1. Family pedigrees could not be prepared due to the disinterest of family members.

### 3.2. Clinical Investigations and Sequence Analyses

Sequencing analysis of *SLC2A9* revealed 12 intron variants (rs2276962, rs2276963, rs2240722, rs2240721, rs2240720, rs28592748, rs16891971, rs3733590, rs13115193, rs4292327, rs6823877, and rs61256984), three synonymous variants (rs13113918, rs10939650, and rs13125646), and four common nonsynonymous allelic variants (rs2276961, p.G25R; rs16890979, p.V282I; rs3733591, p.R294H, and rs2280205, p.P350L). Sequencing analysis of *SLC22A12* revealed three intron variants (rs373881060, rs368284669, and rs11231837), four synonymous (rs3825016, rs11231825, rs1630320, and rs7932775), and two previously identified rare nonsynonymous allelic variants c.1245_1253del (p.L415_G417del) and c.1400C>T (rs200104135, p.T467M), Figure 1. In 14 subjects, we identified one heterozygote and one homozygote for c.1245_1253del, nine heterozygotes and one homozygote for c.1400C>TT, and two compound heterozygotes for c.1400C>T and c.1245_1253del. The genetics data are summarized in Table 2.

In the control group of 109 unrelated subjects of Roma ethnicity from Macedonia, we identified none subject for c.1245_1253del, 10 heterozygotes, and two homozygotes for c.1400C>T.

## 4. Discussion

Renal hypouricemia 1 is a rare inherited disorder, and like other genetic traits and conditions, shows genetic allelic and locus heterogeneity. However, population-specific common variants have been observed in hypouricemia in Japanese, Korean, and Roma patients. In the Japanese [10] and Korean populations [15], the prevalent RHUC1 variants are c.774G>A (p.W258X, frequency 2.30–2.37%) and c.269G>A (p.R90H, frequency 0.40%); in the European Roma population, the prevalent variants are c.1245_1253del (p.L415_G417del, frequency 1.92%) and c.1400C>T (p.T467M, frequency 5.56%). The number of published cases of patients with RHUC1 is about 200, and they are very unlikely to match the frequencies of prevalent variants mentioned above. The key to an early diagnosis is greater awareness of URAT1 deficiency among primary care physicians, nephrologists, and urologists; the recently published clinical practice guidelines for renal hypouricemia will certainly contribute to this [16].

The cohort analyzed in this study was of Roma ethnicity. The Roma represent a transnational ethnic population of 8–10 million, originating in India; currently, they are the largest and most widespread ethnic minority in Europe. In the Slovak Republic, the Roma population is the second-largest minority, with most of the Roma living in the eastern and southern parts of Slovakia. The founder effect and subsequent genetic isolation of the Roma have led to a population specificity, i.e., variants associated with rare diseases in the Roma population tend to be at extremely low frequencies in other European populations and vice versa. Multiple homozygosity for ethnically prevalent mutations is typical in closely related populations with high levels of endogamy, and many rare-disease-causing mutations present in other European populations are found at very low frequencies in the Roma population. Several mutations that cause rare diseases unique to the Roma and have been discovered, e.g., Charcot Marie Tooth disease type 4D and 4G (OMIM #601455 and #605285), congenital cataract facial dysmorphism neuropathy (OMIM #604166), Gitelman syndrome (OMIM #263800), and Galactokinase deficiency (OMIM # 230200) [11,12]. Roma population specificity was confirmed by the findings of comorbidities in our hypouricemic cohort. Of the 27 participants, 5 were diagnosed with phenylketonuria (PKU, OMIM #261600), 4 with short-chain acyl-CoA dehydrogenase deficiency (SCADD, OMIM #201470), 1 with Gilbert syndrome (OMIM #143500), 1 with congenital hypothyroidism (OMIM #275200), and 1 with mild hyperhomocysteinemia due to a methylenetetrahydrofolate reductase deficiency (OMIM #607093.0004). The unusually high prevalence of SCADD deficiency among Roma was confirmed by data from the Slovak newborn screening program in 2016, in which the prevalence of SCADD in Caucasian newborns is 1:9745, while in Roma newborns, it was roughly 1:100 [17].

The first findings indicating population specificity of RHUC1 in the Roma population occurred in 2013, in the context of diagnosing the first RHUC patients in the Czech Republic [18] when two of the three nonconsanguineous probands (compound heterozygotes c.[1245_1253del] + [1400C>T] and homozygotes c.1245_1253del) were from the minority Roma population. Functional studies found significantly reduced urate uptake and a mislocalized URAT1 signal in both variants [14], which were subsequently identified in other patients with hypouricemia of Roma ethnicity in the Czech Republic [19,20]. We subsequently collected a control set of 218 alleles from the Roma population in the Czech Republic and found two heterozygous control subjects with the variant p.L415_G417del. Serum UA and FE-UA in the 9 year old girl was 256 µmol/L and 18%; in the 34 year old man, it was 232 µmol/L and 8%. Further analysis of a cohort of 881 randomly chosen ethnic Roma from two regions in Eastern Slovakia (Prešov and Košice) and two regions in the Czech Republic, identified 25 heterozygous and 3 homozygous subjects for the c.1245_1253del and 92 subjects were heterozygous, 2 were homozygous for the c.1400C>T, and 2 participants were compound heterozygotes. Frequencies of the c.1245_1253del and c.1400C>T variants were determined as 1.87% and 5.56%, respectively [12]. In a subsequent study, we expanded this cohort to include other regions: a total of 1016 Roma from five regions were included: 471 from the Prešov region (Slovakia), 331 from the Košice region (Slovakia), 90 from the Central Bohemia region (Czech Republic), 64 from the Hradec Kralove region (Czech Republic), and 60 from the Iberian Peninsula (Spain). Frequencies of the c.1245_1253del and c.1400C>T variants were determined to be 1.92% and 5.56%, respectively. Genotypes were in Hardy–Weinberg equilibrium (HWE) at the c.1400C>T locus in all five cohorts, but, due to a large number of homozygotes, they were grossly deviated at the c.1245_1253del locus (*p* < 0.001 and *p* = 0.004) in the one Czech and Spanish cohorts. On the other hand, in the control group of 109 nonrelated Macedonia Roma subjects, we did not identify deletion variant c.1245_1253del, while variant c.1400C>T was represented in high frequency of 6.4%. The frequencies of both c.1245_1253del and c.1400C>T variants differ significantly between cohorts. These differences were likely caused by genetic drift, which is a typical phenomenon among Roma subisolates. These findings were subsequently supported by a study of renal hypouricemia in Spanish patients, where eight RHUC1 probands from eight families, all of Roma ethnicity, were identified carrying these prevalent variants (one heterozygote c.1245_1253del, two heterozygotes, four homozygotes c.1400C>T, and one compound heterozygote) [21].

Historic, linguistic, and genetic studies identify India as the original homeland of the Roma. Analysis of paternal and maternal lineages as well as autosomal whole-genome studies date the time of the Roma departure from India to approximately 1000 years ago. The data suggest that the group of Roma who left India had a limited size of around 1000 individuals and came from one specific caste or group [22]. Our findings of the presence of the c.1400C>T variant in all analyzed Roma subcohort, together with data from ExAC database when p.T467M was identified in only one heterozygote subject [12] and three homozygotes in a South Asia cohort of 15,296 subjects but in none of the subjects from all other populations (total 105,759 subjects), suggest the origin of the c.1400C>T variant in Asia. At present, in Genome Aggregation Database (gnomAD), which is the largest and most widely used publicly available collection of population variation, indicates c.1400C>T is present with a global frequency of 0.11% (275 c.1400C>T alleles out of a total of 273,462 sequenced). Worldwide, the highest frequency is present in South Asia (0.70%), while in Europe, it is more common in the southern parts, with the highest frequency among Bulgarians (0.26%). The NGS data from a total of 270 Macedonian patients with different rare diseases, referred for clinical exome analysis at RCGEB-MASA, revealed 5 c.1400C>T heterozygotes (personal communication). All c.1400C>T carriers were of Macedonian ethnic origin. This frequency of 0.93% among Macedonians is the highest reported among the non-Roma populations. The study of the chromosomal background of the c.1400C>T allele could help trace the age/region of origin and the spread of this variant. Taken together, our findings show that the founder effect played a key role in forming the current gene pool of the Roma population — a very small group of migrants from India gave rise to the current population numbering in the low millions. In addition to the primary effect that resulted in the specific gene pool of the Roma ethnic group in general, secondary and even tertiary founder effects played an important role in forming the gene pools of individual Roma subpopulations together with a high degree of consanguinity [22].

The absence of deletion variant c.1245_1253del in two of the Roma subgroups studied may be due to the limitations of the small sample size. However, deletion variant c.1245_1253del has yet to be identified in any of the publicly available databases. These findings suggest the possible origin of the c.1400C>T variants as a founder effect after the departure from India. Taken together, and due to the putative founder effect and genetic drift, the prevalence of this otherwise rare inherited disease is significantly increased in selected populations such as the Roma. In our cohort, most RHUC1 patients also carry the p.G25R variant of GLUT9. However, this allelic variant p.G25R in the GLUT9 transporter, as other identified variants p.V282I, p.R294H, and p.P350L in our clinical cohort, was previously functionally characterized with no influence on expression, subcellular localization, or urate uptake of GLUT9 [23]. Further gene–gene interaction studies are needed to address the possible interaction of URAT1 and GLUT9 variants.

In addition, we can emphasize the clinical importance of documentation regarding RHUC1 in Slovak Roma, which was our focus in this study, given that detailed studies on serum UA and FE-UA levels in Roma are rare. Hypouricemia in children is relatively rare in clinical practice and is often associated with rare conditions. As we showed previously [24], serum UA and FE-UA levels are quite dynamic in the first year of life. Briefly, serum UA levels are low in infancy (131–149 µmol/L at 2–3 months of age) due to high FE-UA levels (>10%); FE-UA levels decrease to approximately 8% by one year of age, where they remains throughout childhood; this is associated with mean serum UA levels of 208–268 µmol/L in children. After age 12, FE-UA levels decrease significantly in boys but not in girls, resulting in a further significant increase in serum UA levels in young men but not in young women. Our data suggest that the high prevalence of URAT1 allelic variants causing impaired urate transport may affect serum UA concentrations in the general Roma population. This hypothesis is supported by clinical studies suggesting that Roma have a 2.85-fold higher risk of end-stage renal disease compared to non-Roma [25] and that Roma women have half-higher odds for nephropathy (OR 1.56; 95% CI 1.01–2.42; *p* < 0.05) than non-Roma women [26]. Moreover, a recent study of Eastern Slovak Roma and non-Roma populations showed that serum UA is ethnicity-specific: the mean UA level ± standard deviation was significantly lower in Roma than in non-Roma (226.54 ± 79.8 µmol/L vs. 259.11 ± 84.53 µmol/L; *p* < 0.0001) [27].

Most patients with RHUC are asymptomatic. However, the prevalence of renal stones due to excesses of UA excretion is 6–7 times higher in patients with RHUC than in individuals with normal uric acid levels [28]. The incidence of urolithiasis, nephrolithiasis, and acute kidney injury among the Roma may be explainable in relation to the population’s risk of RHUC1. To prevent patients from developing complications, physicians often advise patients to drink water before exercise and/or limit the intensity of exercise. Early diagnosis of RHUC has great clinical relevance for correct disease control, thus is important to prevent serious clinical manifestations.

## 5. Conclusions

Generally, the Roma ethnic group represents a cluster of genetically isolated founder populations with a spectrum of frequent Mendelian disorders due to a high rate of consanguinity associated with the occurrence of founder mutations. Therefore, it is necessary to study individual Roma groups to reveal differences in the number of carriers and prevalence of rare diseases in Roma subpopulations. Our data show a high incidence of genetic variants leading to renal hypouricemia 1 among the Roma, and this genetic background should be kept in mind during initial diagnosis.

## Figures and Tables

**Figure 1 biomedicines-09-01607-f001:**
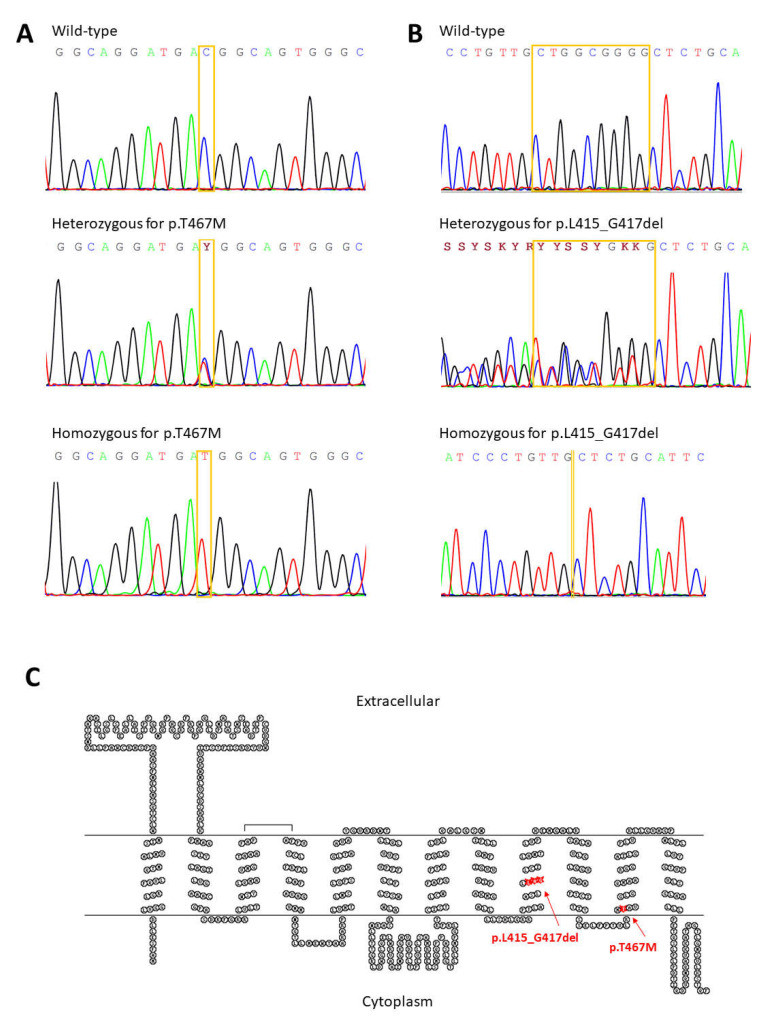
Illustration of allelic variant p.L415:_G417del and p.T467M in genetic and protein context. (**A**) Electropherograms of partial sequences of exon 7 showing the wild-type, heterozygous, and homozygous c.1245_1253del variant. (**B**) Electropherograms of partial sequences of exon 9 showing wild-type, heterozygous, and homozygous c.1400C>T variant. (**C**) Position of identified allelic variant p.T467M and p.L415_G417del in a URAT1 membrane topology model created via Johns S.J., TOPO2, Transmembrane protein display software, http://www.sacs.ucsf.edu/TOPO2/ [23 July 2021].

**Table 1 biomedicines-09-01607-t001:** The clinical and biochemical data of the analyzed cohort.

Family	Patient	Sex	Year of Birth	Year of Onset	sUA µmol/L	FE-UA %	Associated Diseases
A	1	F	1962	N/A	263	7.6	PKU heterozygote
	2	F	1991	N/A	137	9.7	PKU
B	3	M	2014	4	101 (2018), 102 (2021)	13.1 (2018), 16.2 (2021)	autism, ADHD
	4	M	1985	N/A	326	7.3	
	5	F	1981	20	167	8.6	
C	6	M	2006	12	94 (2018)	23.4 (2018)	PKU
	7	M	2012	6	122 (2017), 150 (2018)	15.8 (2018)	PKU
	8	F	2011	7	118 (2018)	12.4 (2018)	PKU
D	9	F	2008	10	194 (2018)	7.3 (2018)	MTHFR homozygote, PAI heterozygote
	10	F	1999	20	141 (2019)	16.2 (2019)	SCADD heterozygote
E	11	F	2018	10 mth	95 (2018), 146 (2019), 151 (2021)	17.1 (2018), 12.0 (2021)	SCADD
	12	F	1994	24	N/A	N/A	SCADD heterozygote
F	13	F	1991	N/A	309 (2018)	5.1 (2018)	xanthinuria heterozygote
	14	M	2017	18 mth	72 (2018), 136 (2019), 141 (2021)	10.3 (2021)	PKU
	15	M	2016	3	135 (2016), 140 (2019)	11.3 (2019)	SCADD
G	16	M	2018	6 mth	140 (2018), 114 (2019), 235 (2020)	16.5 (2018)	transitory hypertyrosinemia
	17	F	1981	38	247 (2018)	8.4 (2018)	
	18	M	1988	31	415 (2018)	5.9 (2018)	
H	19	M	2008	11	55 (2019), 77 (2020)	28.8 (2020)	hypogonadotropic hypogonadism, Gilbert syndrome, hepatopathy, cholecystolithiasis, obesity
	20	F	1986	34	125	11.6	
	21	M	1983	37	44	43	
I	22	F	2016	27 mth	118 (2017), 131 (2019), 152 (2021)	10.6 (2019)	central hypocorticism, epilepsy, familiar hypercholesterolemia
	23	F	1982	37	161	8.2	neurofibromatosis type 1
	24	M	1984	35	366	6.1	familial hypercholesterolemia
J	25	M	2019	23 mth	61 (2019), 43 (2020)	51 (2019)	SCADD
	26	F	2002	18	N/A	N/A	SCADD heterozygote
K	27	M	2010	10	108 (2012), 88 (2013), 84 (2018), 71 (2019), 49 (2021)	56 (2012), 30 (2013), 40 (2018), 40 (2019), 50 (2021)	SCADD, prematurity, perinatal hypoxia, psychom. retardation, cardiomyopathy

Reference ranges are as follows. SUA: 120–360 µmol/L (2.02–6.05 mg/dL) (< 15 years and female), 120–420 µmol/L (2.02–7.06 mg/dL; male); FE-UA: reference range 5–20% (< 13 years), 5–12% (male) and 5–15% (female). PKU, phenylketonuria; SCADD, short-chain acyl-CoA dehydrogenase deficiency; MTHFR, methylenetetrahydrofolate; PAI- 1, plasminogen activator inhibitor-1; ADHD, attention deficit hyperactivity disorder; mth, months; N/A, not available.

**Table 2 biomedicines-09-01607-t002:** The genetic data of the analyzed cohort identified nonsynonymous allelic variants in URAT1 and GLUT9 transporter; homozygous variants are shown in bold font; causal variants for RHUC1 are in *SLC22A12* gene.

Family	Patient	Gene *SLC22A12,* URAT1	Gene *SLC2A9,* GLUT9
A	1	reference	N/A
	2	reference	p.G25R, **p.V282I**, p.P350L
B	3	reference	p.G25R, p.V282I
	4	reference	p.G25R, p.V282I, p.R294H
	5	p.T467M	N/A
C	6	p.T467M	p.G25R, p.V282I, p.R294H
	7	p.T467M	p.G25R, p.V282I, p.R294H
	8	p.T467M	**p.G25R**, p.R294H
D	9	reference	**p.G25R**, p.R294H, p.P350L
	10	p.T467M	**p.G25R**, p.R294H, p.P350L
E	11	p.T467M	p.G25R, p.V282I
	12	reference	p.G25R
F	13	reference	p.G25R, p.V282I, **p.P350L**
	14	p.T467M	p.G25R, p.V282I, p.P350L
	15	p.L415_G417del	p.G25R, p.P350L
G	16	reference	N/A
	17	reference	p.P350L
	18	reference	**p.G25R**, p.R294H, p.P350L
H	19	p.L415_G417del, p.T467M	**p.R294H, p.P350L**
	20	p.T467M	N/A
	21	**p.L415_G417del**	N/A
I	22	reference	**p.G25R**, p.V282I, p.P350L
	23	reference	p.G25R, p.V282I, p.P350L
	24	reference	**p.G25R**, p.R294H, p.V282I, p.P350L
J	25	p.L415_G417del, p.T467M	**p.G25R**, p.R294H, p.P350L
	26	p.T467M	N/A
K	27	**p.T467M**	N/A

## Data Availability

The data presented in this study are available on request from the corresponding author.

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
