# Peer review of "Renal Hypouricemia 1: Rare Disorder as Common Disease in Eastern Slovakia Roma Population"

_biomedicines, 2021, doi:10.3390/biomedicines9111607_

Round 1

Reviewer 1 Report

This study reported the genetic characterization of the renal hypouricemia (N=27 from 11 family) identified from the cohort of the Roma population. However, the study has several issues, especially, about the lack of novelty and unknown prevalence of the mutation in the population.

  1. Similar topic was already reported previously (PMID: 26033041, and 27906637). These studies have shown that URAT1 mutation causing hypouricemia is not rare in the Roma population. To show novelty of the study, more detailed analysis would be required. At least, the prevalence of the disease and causative mutation in the population need to be estimated. Otherwise, the present study is just a case series of renal hypouricemia in the specific population.
  2. I could not find the detail information of the cohort characteristics. It needs to describe and/or refer the previous paper.
  3. We could not follow how the patients were screened. Was it done by regular examination in the asymptomatic patients or symptomatic patients?

4 This study did not mention how many individuals were examined as population parameter. Thus, it is hard to estimate the prevalence of renal hypouricemia and URAT1 mutation in the cohort.

  1. The authors need to describe each mutation is novel or reported ones. If they are novel mutation, no functional study or no prediction analysis of the structural effect cannot conclude them as the causative mutation.
  2. In table 2, some subjects possess mutations both in URAT1 and GLUT9. This table is difficult to understand what mutation is the pathologic mutation for hypouricemia. It needs to revise the table. Some patients (e.g. E12, G17 etc) possesses only heterozygous GLUT1 mutation. Why can these mutation cause hypouricemia despite heterozygous mutation? I recommend to add family tree information in each case.

7 The examined families show high frequency of comorbidity of other genetic diseases, raising the possibility that some of the families are consanguineous. If so, it would make difficult to estimate the actual prevalence of the disease/mutation in the general population.

8, In the Japanese population, the pathologic mutations of hypouricemia are widely located in the gene of URAT1 (PMID 32005656). However, in the present study, the mutations are localized in the specific cite. Is this due to the high frequency of consanguineous in the cohort? Please discuss.  

9 Ethical issues. How did you obtain the consent from the subjects of child? The ethical method was not described. In table, to address year of birth looks meaningless, rather it generates ethical problem for identification of the individual.

10 The issue of term. Renal hypouricemia does not cause not general acute kidney injury, but can cause exercise-indued acute kidney injury.

Author Response

Dear Editors

Re: biomedicines- 1402355

Title: Renal hypouricemia 1: rare disorder as common disease in Eastern Slovakia Roma population

Authors: Blanka Stiburkova, Jana Bohatá, Kateřina Pavelcová, Velibor Tasic, Dijana Plaseska-Karanfilska, Sung Kweon Cho, Ludmila Potočnaková, Jana Šaligová

Thank you for your letter and for the positive and helpful comments of the reviewers. We have made revisions to the manuscript in response to the comments. The changes are described below.

Reviewer reports:

Reviewer #1: Round 1

This study reported the genetic characterization of the renal hypouricemia (N=27 from 11 family) identified from the cohort of the Roma population. However, the study has several issues, especially, about the lack of novelty and unknown prevalence of the mutation in the population.

  1. Similar topic was already reported previously (PMID: 26033041, and 27906637). These studies have shown that URAT1 mutation causing hypouricemia is not rare in the Roma population. To show novelty of the study, more detailed analysis would be required. At least, the prevalence of the disease and causative mutation in the population need to be estimated. Otherwise, the present study is just a case series of renal hypouricemia in the specific population.

We agree with the reviewer that we have described the URAT1variants in our previous papers. However, in this manuscript, is a priority communication from clinical practice that highlights the underdiagnosis of RHUC and the need to include data on Roma ethnicity in the diagnostic algorithm. We present a heterogeneous cohort of patients recruited during the study period from patients regularly monitored for various diseases in the metabolic outpatient clinic of the Children's University Hospital in Košice. The majority of the Slovak Roma population lives in the eastern Slovak region of Košice. The reasons for the extension of the diagnosis of RHUC and inclusion of the patient in the cohort were recurrent hypouricemia and hyperuricosuria (not caused by the primary previously established diagnosis) and Roma ethnicity.  As published in the mentioned and other studies, an unusually high frequency of various rare inherited diseases was found in the Slovak Roma population. Given the higher prevalence of nephropathy presented in the Slovak Roma population (references 24,25) together with the high prevalence of dysfunctional URAT 1 variants (references 11,12,14,15), we hypothesized and confirmed that RHUC is underdiagnosed.

Our findings illustrate why genetic prevalent variants affecting URAT1 function should be routinely considered in clinical practice in the evaluation of patients with hypouricemia and hyperuricosuria presenting with clinical symptoms such as urolithiasis, nephrolithiasis, and acute kidney injury, namely in patients of Roma ethnicity.

Moreover, the strength of our study is to investigate the role of founder mutation (c.1400C>T (p.T467M) in a different Roma cohort. The article is enriched with a first-time analyzed cohort of Macedonian Roma, comprising 109 unrelated persons. The data confirmed the prevalence of the c.1400C>T variant, which was represented with a high frequency of 6.4%. On the other hand, variant c.1245_1253del was not detected, which, together with the available data from the databases, points to a different mechanism and timing of the prevalent Roma variants for rare hereditary hypouricemia type 1.

Taken together, we suppose that 1,125 randomly selected ethnic Roma from six regions/four countries is a sufficient sample to estimate the prevalence of population-specific variants.

  1. I could not find the detail information of the cohort characteristics. It needs to describe and/or refer the previous paper.

The presented cohort was recruited within a study period from the patients and their family members followed up for various diseases or examined because of differential diagnostic reasons in the Metabolic ambulance of Children´s Faculty Hospital in Kosice. As it is explained later, repeated hypouricemia was crucial to the inclusion into the cohort. The characteristic of the cohort is summarized in Table 1. The paragraph about the control group was add-in the section of Method.

  1. We could not follow how the patients were screened. Was it done by regular examination in the asymptomatic patients or symptomatic patients?

Thanks for your suggestion and we are sorry for the unclear description. As it was previously written the presented cohort was recruited within a study period from the patients followed up for various diseases or examined because of differential diagnostic reasons in the Metabolic ambulance of Children´s Faculty Hospital in Kosice, which means not primarily because of the hypouricemia.  Analysis of the uric acid is a part of a differential diagnostic analysis panel. Examination of the family members belongs to the examination algorithm. Hypouricemia was a coincidental finding in probands. Repeated hypouricemia and hyperuricosuria (not caused by the previously proven primary mainly inherited diagnosis) and Roma ethnicity were the reason for the diagnostics extension of RHUC and inclusion of the patient in the cohort. These mentioned not explained laboratory findings were not present within the study period in patients of the majority Slovak population.

  1. This study did not mention how many individuals were examined as population parameter. Thus, it is hard to estimate the prevalence of renal hypouricemia and URAT1 mutation in the cohort.

The Metabolic ambulance of Children´s Faculty Hospital in Kosice is one of the three metabolic ambulances in Slovakia. It provides the healthcare service for the East Slovakia region. The presented cohort was recruited from the all patients followed up or examined within a study period, independently of ethnicity, it means from both populations - majority and minority-Roma population. As it was previously written these findings were found only in patients of Roma ethnicity. No one with these findings from the majority population was found within the study period in spite of the fact that the number of patients from majority populations is much higher (and also earlier and later- a personal experience of the co- author- this information is not described in the actual article). That is why we suppose a much higher prevalence in the Roma population than in the majority one.

Total as a cohort to estimate the prevalence of population-specific variants in the common Roma population we analyzed 1,125 randomly selected ethnic Roma subjects from six regions.

  1. The authors need to describe each mutation is novel or reported ones. If they are novel mutation, no functional study or no prediction analysis of the structural effect cannot conclude them as the causative mutation.

We have previously functionally characterized the variants we detected in both URAT1 and GLUT9 as described in the Discussion.

Functional studies found significantly reduced urate uptake and a mislocalized URAT1 signal in both variants p.L415_G417del, and c.1400C>T (13).

The allelic variant p.G25R in the GLUT9 transporter, as others identified variants p.V282I, p.R294H, and p.P350L in our clinical cohort, was previously functionally characterized with no influence on expression, subcellular localization, and urate uptake of GLUT9 (23).

  1. In table 2, some subjects possess mutations both in URAT1 and GLUT9. This table is difficult to understand what mutation is the pathologic mutation for hypouricemia. It needs to revise the table. Some patients (e.g. E12, G17 etc) possesses only heterozygous GLUT1 mutation. Why can these mutation cause hypouricemia despite heterozygous mutation? I recommend to add family tree information in each case.

Thanks for your suggestion and we are sorry for the unclear description. We reviewed the table and clearly identified the causative mutations for renal hypouricemia.

Biochemical and clinical markers for RHUC1 and RHUC 2 are similar. Thus, we analyzed both causative genes SLC22A12 and SLC2A9. Variants identified in the SLC2A9 gene have no causal association with RHUC and have been previously functionally characterized as neutral. On the other hand, only variants causal for RHUC1 were found in the SLC22A12 gene.

We completely agree with the opinion that it would be advisable to have pedigrees for all the families described. Unfortunately, this is not possible due to the lack of interest of family members; we have not been able to obtain the consent to enumerate the entire family or the first-degree relatives. We have extended the method paragraph for this omission.

  1. The examined families show high frequency of comorbidity of other genetic diseases, raising the possibility that some of the families are consanguineous. If so, it would make difficult to estimate the actual prevalence of the disease/mutation in the general population.

All the examined families were Slovaks of Roma ethnicity (except for family 1, who did not report their ethnicity). The present Roma population came from the small group of a few hundred families who migrated to Europe from Asia in the 13th century. Roma population lived centuries isolated without mixing with the majority population.  Many rare genetic diseases caused by the single mutation typical for the Roma population (practically nor found or very rare in a majority population)  were described as a result of multiple are founder effect (it means without anamnesis of the known consanguineous relationship).

The families under investigation did not indicate a family relationship. In our revised manuscript, we have added this sentence and apologize for the omission.

  1. In the Japanese population, the pathologic mutations of hypouricemia are widely located in the gene of URAT1 (PMID 32005656). However, in the present study, the mutations are localized in the specific cite. Is this due to the high frequency of consanguineous in the cohort? Please discuss.

Thanks for your suggestion and we are sorry for the unclear description. We have added this part to the discussion section of the manuscript: Taken together, our findings showed that the founder effect played a key role in forming the current gene pool of the Roma population — a very small group of migrants from India gave rise to the current population numbering in the low millions. In addition to the primary effect that resulted in the specific gene pool of the Roma ethnic group in general, secondary and even tertiary founder effects played an important role in forming the gene pools of individual Roma subpopulations together with a high degree of consanguinity (22).

  1. Ethical issues. How did you obtain the consent from the subjects of child? The ethical method was not described. In table, to address year of birth looks meaningless, rather it generates ethical problem for identification of the individual.

All participants were fully informed of the study’s goals and written informed consent was obtained from each participant before enrollment in the study. According to the law of the country, the caregiver of the child (usually its parent or the other legal caregiver) is responsible to sign the consent with the examination. All tests were performed following standards set by the institutional ethics committees. The study was approved on 24 July 2018 as part of project no. 7131/2018. All the procedures were performed per the Declaration of Helsinki.

Year of birth is included in the table because of the dynamic change in FE-UA in children. As can be seen in the table, both major markers of RHUC (s-UA, FE-UA) change over time, especially in preschool children.

  1. The issue of term. Renal hypouricemia does not cause not general acute kidney injury, but can cause exercise-indued acute kidney injury.

As it is written in the article most patients with RHUC are asymptomatic. However, the prevalence of renal stones due to excesses of UA excretion is 6–7 times higher in patients with RHUC than in individuals with normal uric acid levels (27). The incidence of urolithiasis, nephrolithiasis, and acute kidney injury among the Roma may be explainable in relation to the population´s risk for RHUC 1.

Proper hydration during physiological activity advised by physicians (as exercise) is also according to authors meaning, reasonable and easy to do advice for the patients to prevent developing renal complications.

Reviewer 2 Report

The manuscript by Blanka Stiburkova et al "Renal hypouricemia 1: rare disorder as common disease in Eastern Slovakia Roma population" reports clinical, biochemical and genetic findings in a cohort recruited from the Košice region of Slovakia consisting of subjects with hypouricemia and Roma ethnic relatives.

The manuscript describes very credibly the baseline, population presentation and genetic analysis indicating frequent dysfunctional variants of URAT1 in Roma ethnicity.

I believe that this is an appropriately structured article with a discussion that does not need to be fundamentally altered prior to publication.

Author Response

We are very grateful for your comments on the manuscript.

Round 2

Reviewer 1 Report

The authors have responded appropriately to my commments. 

This manuscript is a resubmission of an earlier submission. The following is a list of the peer review reports and author responses from that submission.

Round 1

Reviewer 1 Report

This is a study of 27 participants, 9 hypouricemia probands
and 18 family relatives. However, 12 of them (44%) had other inherited diseases,
and many of them had associated diseases from autism to hepatopathy,
cholecystolithiasis, epilepsy, retardation, cardiomyopathy, etc.
I think this makes the cohort very heterogeneous and
there is always the
question of whether and to what extent these inherited diseases and
associated diseases have affected the results.
Furthermore, the study lacks
healthy control of the Roma population and/or perhaps other ethnic groups,
in order to conclude that these high incidences of genetic variants
are only or predominantly associated with this population.

Author Response

We completely agree with the reviewer that this is a heterogeneous cohort of patients from clinical practice - the presented cohort was recruited within a two-year study period 2018 – 2020 from the patients followed up for various diseases in the Metabolic ambulance of Children´s Faculty Hospital in Kosice. Repeated hypouricemia and hyperuricosuria (not caused by the previously proven primary mainly inherited diagnosis) and Roma ethnicity were the reason for the diagnostics extension of RHUC and inclusion of the patient in the cohort. These mentioned not explained laboratory findings were not present within the study period in patients of the majority Slovak population.

The genetic variants of RHUC in the Roma and other populations were already published by the first author in the previous studies (references 8,9,11,12,13,14,15). In the largest study, a total of 1,016 Roma subjects from the general population (without biochemical data) from five regions were included: 471 from the Prešov region (Slovakia), 331 from the Košice region (Slovakia), 90 from the Central Bohemia region (Czech Republic), 64 from the Hradec Kralove region (Czech Republic), and 60 from the Iberian Peninsula (Spain). Frequencies of the c.1245_1253del and c.1400C>T (p.T467M) variants were determined 1.92% and 5.56%, respectively.

On the other hand, in the database ExAc variant p.T467M was identified in only three homozygotes subjects in a South-Asia cohort of 15,296 but in none of the subjects from European and other populations (total 105,759 subjects). Moreover, the deletion variant p.L415_G417 has yet to be identified in any of the publicly available databases indicating a high population specificity for the Roma population.

The clinical studies suggesting that Roma have a 2.85-fold higher risk of end-stage renal disease compared to non-Roma (references 24) and that Roma women are half-higher odds for nephropathy (OR 1.56; 95% CI 1.01-2.42; p < 0.05) than non-Roma women (references 25). Moreover, a recent study of Eastern Slovak Roma and non-Roma population showed that serum UA is ethnicity-specific: the mean UA level ± standard deviation was significantly lower in Roma than in non-Roma (226.54 ± 79.8 µmol/l vs. 259.11 ± 84.53 µmol/l; p < 0.0001) (references 26).

Taken together, our data showed the high incidence of prevalent variants of renal hypouricemia 1 among the Roma in comparison with other populations. This genetic background should be kept in mind during a diagnosis scheme of patients with hypouricemia/hyperuricosuria presenting with clinical symptoms such as urolithiasis, nephrolithiasis, and acute kidney injury, namely in patients of Roma ethnicity.

Reviewer 2 Report

The authors evaluated “Renal hypouricemia type 1 in Eastern Slovakia Roma population”. The sarcopenia is emerging and important issue in PD patients. The paper showed that hypouricemic Roma patients have high prevalence of mutation of c.1245_1253del and/or c.1400C>T in SLC22A121 gene. The study is very interesting, but the some researches regarding same mutation was published by same authors. Originality for mutation per se may be low. In addition, there was very low sample size to present high prevalence of the relevant muation in Roma population. Although the study included different patients using Roma population, the results or conclusion may present sufficient originality compared to previous studies.

Previous three studies by same mutation and/or similar ethnicity

1. Prevalence of URAT1allelic variants in the Roma population.

Stiburkova B, Gabrikova D, Čepek P, Šimek P, Kristian P, Cordoba-Lanus E, Claverie-Martin F.Nucleosides Nucleotides Nucleic Acids. 2016 Dec;35(10-12):529-535

2. A heterozygous variant in the SLC22A12 gene in a Sri Lanka family associated with mild renal hypouricemia.

Vidanapathirana DM, Jayasena S, Jasinge E, Stiburkova B.BMC Pediatr. 2018 Jun 29;18(1):210.

3. High frequency of SLC22A12 variants causing renal hypouricemia 1 in the Czech and Slovak Roma population; simple and rapid detection method by allele-specific polymerase chain reaction.

Gabrikova D, Bernasovska J, Sokolova J, Stiburkova B.Urolithiasis. 2015 Oct;43(5):441-5.

Other minor comment

In Figure 1C, the figure is similar to a figure from previous paper (JASN, 2004;15(1):164-173). Although the author revised the figure by presentation of different mutant points, the back-bone of the figure may be same. If the figure was referenced by the previous study, the figure may be approved by the relevant journal or authors.

Author Response

We agree with the reviewer that we have described the published variants in our previous papers. Studies 1 and 3 were performed on Roma subjects from the general population, unfortunately without the availability of biochemical data. Study 2 demonstrated a mild clinical form of RHUC 1 and detected the p.T467M variant outside the Roma population for the first time: the p.T467M variant was identified in only one heterozygote (12) in a South Asian cohort of 15 296 individuals but in none of the individuals from all other populations (105 759 individuals in total).

However, in this manuscript, this is a priority communication from clinical practice that highlights the underdiagnosis of RHUC and the need to include data on Roma ethnicity in the diagnostic algorithm.

In this study, we present a heterogeneous cohort of patients recruited during the study period from patients regularly monitored for various diseases in the metabolic outpatient clinic of the Children's University Hospital in Košice. The majority of the Slovak Roma population lives in the eastern Slovak region of Košice (published by the authors Dluholucky S, Knapková M. from the Newborn Screening Centre of Slovak Republic: first results of extended newborn screening in Slovakia - differences between the majority and Roma ethnicity in reference 18). The reasons for the extension of the diagnosis of RHUC and inclusion of the patient in the cohort were recurrent hypouricemia and hyperuricosuria (not caused by the primary previously established diagnosis) and Roma ethnicity.  As published in the mentioned and other studies, an unusually high frequency of various rare inherited diseases was found in the Slovak Roma population. Given the higher prevalence of nephropathy presented in the Slovak Roma population (references 24,25) together with the high prevalence of dysfunctional URAT 1 variants (references 11,12,14,15), we hypothesized and confirmed that RHUC is underdiagnosed.

Our findings illustrate why genetic prevalent variants affecting URAT1 function should be routinely considered in clinical practice in the evaluation of patients with hypouricemia and hyperuricosuria presenting with clinical symptoms such as urolithiasis, nephrolithiasis, and acute kidney injury, namely in patients of Roma ethnicity.

Figure 1

We apologize for the inadvertent omission, the image has been revised.

Round 2

Reviewer 2 Report

Thank you for response of comments. We agree the author's comments. Although renal hypouricemia is rare disease, clinical course is also well-known. The study for this rare disease should include data for new genetic mutation, large sample size, or uncommon presentation of disease course. Although the study is very interesting, this study may not overcome these limitations.